# Italian Translation, Cultural Adaptation, and Validation of the Toileting Habit Profile Questionnaire Revised (THPQ-R) in Typically Developing Children: A Cross-Sectional Study

**DOI:** 10.3390/children9071052

**Published:** 2022-07-14

**Authors:** Martina Ruffini, Anna Berardi, Anna Benvenuti, Isabelle Beaudry-Bellefeuille, Marco Tofani, Giovanni Galeoto, Donatella Valente

**Affiliations:** 1Department of Human Neurosciences, School of Occupational Therapy, Sapienza University of Rome, 00185 Rome, Italy; ruffinimartina@gmail.com (M.R.); anna.benvenuti.10@gmail.com (A.B.); 2Department of Human Neurosciences, Sapienza University of Rome, 00185 Rome, Italy; marco.tofani@uniroma1.it (M.T.); giovanni.galeoto@uniroma1.it (G.G.); donatella.valente@uniroma1.it (D.V.); 3Director of Clinica De Terapia Ocupational Pediatrica Beaudry-Bellefeuille, 33007 Oviedo, Spain; ibbergo@gmail.com; 4IRCSS Neuromed, Via Atinense, 18, 86077 Pozzilli, Italy

**Keywords:** assessment, defecation, pediatric, rehabilitation, sensory integration

## Abstract

The Toileting Habit Profile Questionnaire Revised (THPQ-R) identifies sensory integration issues in children with defecation problems. Sensory integration issues are recognized as a factor linked to some defecation disorders and identifying such issues is needed to guide the development of an appropriate intervention. The aim of this cross-sectional study was to translate, culturally adapt, and validate the THPQ-R in a population of typically developing (TD) Italian children by measuring its internal consistency and cross-cultural validity. The translation and cultural adaptation were performed according to international guidelines. The questionnaire was administered to the caregivers of TD children, and the analysis was performed on data obtained from 118 TD children. The construct validity of the scale was calculated through the exploratory factor analysis that indicated two factors; Cronbach’s alpha was calculated for internal consistency and showed a value >0.7, demonstrating good internal consistency. Cross-cultural validity was also analyzed and showed higher levels of defecation problems at age 5 years. Italian occupational therapists now have a tool to assess possible sensory issues linked to defecation disorders in children aged 3 to 6 years, which may promote more effective clinical practice in this area. Moreover, it will be possible to compare the results obtained from studies conducted in Italy with those conducted in other countries.

## 1. Introduction

Defecation disorders may be related to sensory deficits and strongly impact an individual’s autonomy and well-being. The acquisition of voluntary bowel continence and autonomy in personal hygiene are considered important milestones in childhood and are therefore among the participation concerns of interest in occupational therapy. Bowel control and personal hygiene are important activities of daily living and problems in these areas can limit an individual’s independence, well-being, and social participation [1,2]. Between the ages of 3 and 6 years, children acquire fecal continence and can safely transition to using a potty or toilet for evacuation. However, this process can be accompanied by conditions that make the transition slower and more difficult. Possible underlying conditions include sensory integration deficits and constipation. These problems can lead to fecal retention, stool toileting refusal, and fecal incontinence. In some cases, pharmacological treatment from a pediatrician or gastroenterologist, along with occupational therapy to promote adequate toileting habits, may be necessary to help the child.

Several salient aspects have emerged from the literature. First, the incidence of problems related to defecation is very high during childhood with a pooled prevalence of 9.5% [3,4]. In particular, children with autism spectrum disorder or attention deficit hyperactivity disorder show a higher prevalence of functional defecation disorders (FDDs) [5,6]. Moreover, children with these concomitant diagnoses tend to have poorer responses to current medical treatments for their defecation disorder [7] and are known to have a higher incidence of sensory integration problems. Therefore, focusing on sensory integration issues, which may impact defecation in children with these disorders, in both research and clinical practice is necessary to develop more effective interventions. Sensory integration dysfunction and defecation disorders are known to affect family occupations and quality of life [8,9,10,11] and therefore warrant the attention of healthcare professionals [12]. Furthermore, the literature highlights a relationship between defecation disorders and sensory integration deficits, or a diagnosis in which sensory integration deficits commonly occur. In particular, sensory hyper-reactivity has been associated with a higher frequency of challenging defecation behaviors [9,13,14]. The results of these studies highlight the importance of screening for sensory hyper-reactivity in clinical practice when working with children with defecation issues. Finally, there are reports that pharmacological treatment combined with occupational therapy using a sensory integration approach can improve the acquisition of age-appropriate toileting habits [9].

The Toileting Habit Profile Questionnaire Revised (THPQ-R) [15,16] was designed to screen for sensory integration problems related to defecation disorders. Both sensory re-activity and perception are components of sensory integration and serve as a foundation for participation in daily life tasks. Reactivity refers to the response and level of comfort relative to the sensation. Perception refers to the interpretation or decoding of the sensory information. Hyporeactivity (under-responsive or unaware of the sensation) and poor perception can be difficult to distinguish, and there is no consensus on how to clearly separate these issues based exclusively on descriptions of behavior. Relative to defecation, adequately perceiving and reacting to the sensations involved in the evacuation of feces is necessary to transition from diaper to toilet or potty. Issues in sensory integration related to defecation include: (1) Poor perception/hyporeactivity: the child does not feel the need to evacuate, or feels it in extremis when there is no longer time to travel to the bathroom, or the child may not realize they are soiled; (2) hyper-reactivity: the child experiences the use of the toilet and defecation per se as an unpleasant sensation and opts to retain stool to avoid defecation or refuses to defecate in the toilet. Both aspects of sensory integration are reflected in the THPQ-R.

The THPQ-R was developed by an occupational therapist in collaboration with gastroenterologists who noticed that many children did not respond to conventional medical and complementary behavioral approaches [15]. In most of the children who had not responded to first-line medical treatment, sensory integration issues were observed [6]. Thus, a link between fecal retention or constipation and the presence of sensory integration disorders emerged and resulted in the development of the THPQ. The THPQ was revised into the THPQ-R, which is considered a valid tool to assess sensory hyper-reactivity in children with defecation problems. English and Spanish versions [17] have been validated and show good construct validity and a good discriminatory capacity between children with FDDs and those without FDDs. Furthermore, this is the only validated tool that evaluates sensory reactivity related to defecation disorders in clinical practice, thus filling a gap in this field. Other instruments used in the evaluation of sensory reactivity issues do not specifically contemplate defecation and toileting sensations. Therefore, for children with FDDs, the use of the THPQ-R in conjunction with other more general sensory questionnaires and evaluation instruments allows therapists to carry out a comprehensive assessment of sensory issues, including those linked to toileting. The aim of this cross-sectional study was to translate, culturally adapt, and validate the THPQ-R in a population of typically developing (TD) Italian children by measuring internal consistency and cross-cultural validity.

## 2. Methods

The developers of the original THPQ-R agreed to translate and culturally adapt the tool into Italian. Thus, the original THPQ-R was translated from English to Italian by the Associazione Italiana Integrazione Sensoriale (SENSIS) using the Translation and Cultural Adaptation of Patient Reported Outcome Measures–Principles of Good Practice guidelines [18].

### 2.1. Translation and Cultural Adaptation

The original THPQ-R was independently translated into Italian by a panel of two native English speakers and one Italian clinical psychologist familiar with English [19,20,21,22,23,24,25]. A native speaker of the target language who had not been involved in any of the forward translations synthesized the results and created a preliminary Italian version of the questionnaire. Working with this version, three Italian translators then translated the questionnaire back into the English without having seen the original version. The back-translated version of the instrument was compared by the team with the original. This version was sent to the original authors of the questionnaire and approved by them. To adapt the translated version to Italian culture, a panel of occupational therapists experienced with sensory integration who were familiar with both languages reviewed the preliminary translated version and produced the final version. The final version was shown to five parents (children between 3 and 8 years old) who were not familiar with sensory integration terminology to check understandability.

### 2.2. Participants and Procedures

Parents of TD children between 3 and 8 years old were recruited to the study through Italian primary schools and online forums. Children with neurodevelopmental, gastrointestinal, or neurological disorders were excluded. The questionnaire was completed using an online form (developed with Google modules [Mountain View, CA, USA]) or a paper form, to which a personal data section regarding the child was added to collect information on age and sex. An adaptation of the instrument and its instructions for the online format was not required because both the paper and the online forms were self-administered; this guaranteed having no bias in validation of the tool between the two formats. Parents interested in participating were informed about the modalities and purposes of the study and signed an informed consent form [26].

### 2.3. Instruments

The first version of the THPQ [14] used a five-point response scale and included 17 items divided into two sections: hyper-reactivity (15 items) and hyporeactivity (2 items). Changes were made to the questionnaire following studies to verify the validity of the construct (Rash analysis). The most recent version of the THPQ, the THPQ-R [16], is composed of 17 items and the scale is now dichotomous, with two response options for each item: (1) frequently or always, or (2) never or rarely, based on the frequency with which the behavior described in the item occurs. Items 16 and 17 are not included in the calculation of the final score because they were shown to represent a different construct, sensory hypo-reactivity or difficulties in perception, a construct that is difficult to assess with questionnaires (9). These items are therefore not considered in the score but are still qualitatively considered within the occupational therapy clinical assessment.

### 2.4. Statistical Analysis

Following the Consensus-Based Standards for the Selection of Health Status Measurement Instrument (COSMIN) checklist, the construct validity, internal consistency, and cross-cultural validity of the scale were assessed [27]. The construct validity of the scale was calculated through the exploratory factor analysis; the Kaiser–Meyer–Olkin value was calculated; Bartlett’s sphericity test was performed. The internal consistency of the THPQ-R was examined by Cronbach’s alpha. Alpha values of 0.7, 0.8, and 0.9 are thought to represent a fair, good, and excellent degree of internal consistency, respectively [28]. Cross-cultural validity was measured using a box plot to highlight the symmetrical or asymmetrical distribution of the results between sex and age. The boxplot is a statistical representation of correlations between the means; we expected to observe a difference in score between ages, specifically lower scores around 5 years of age. A *p* value less than or equal to 0.05 was considered statistically significant for all the statistical analyses. The IBM^®^ SPSS^®^ tool (Chicago, IL, USA) was used to carry out the statistical analysis.

## 3. Results

### 3.1. Translation and Cultural Adaptation

The original version of the THPQ-R was translated and approved by the developers of the tool. After a consensus meeting, the Italian version of the THPQ-R was developed (see Appendix A). The translation committee considered all items to be either identical or similar in meaning to the original English version. A group of five people of the target population (parents of children between 3 and 8 years old) confirmed the understandability of the questionnaire.

### 3.2. Participants

From March 2021, 90 parents of 118 children who met the inclusion criteria participated in the present study. Table 1 summarizes the demographic characteristics of the participants.

### 3.3. Statistical Analysis

To evaluate the construct validity of the THPQ-R, the exploratory factor analysis was used (EFA). The Kaiser–Meyer–Olkin statistic was 0.75, and Bartlett’s test of sphericity was statistically significant (*p* < 0.001). The EFA extracted 17 items that explained 64% of the variance, which was contained in two major factors. Factor 1 included 15 items, while factor 2 included 2 items (Table 2).

As reported in Table 3, the value of Cronbach’s alpha was 0.76 for sensory type 1 (sensory over-reactivity) and 0.75 for sensory type 2 (sensory under-reactivity), demonstrating a good internal consistency, i.e., an excellent inter-relation between items. All items were shown to be relevant and contribute to the questionnaire, given that if any item was eliminated, the alpha value decreased, consequently decreasing the internal consistency. It was also possible to make a correlation between child demographic characteristics and THPQ-R scores using box plots (Figure 1).

## 4. Discussion

This study was conducted by a group of researchers at the Sapienza University of Rome experienced in validating outcome measures, in collaboration with the original developer of the tool [29,30,31,32,33,34]. The goal of this study was to translate, culturally adapt, and validate the THPQ-R in an Italian population of TD children.

From the literature review, we noticed that this is the first study to perform the exploratory factor analysis (EFA); it was not performed even in the original paper for reduction to the revised version. These results indicated sampling adequacy for principal component analysis. The consequent EFA showed that the first fifteen elements of THPQ-R were all included in a single factor. This result confirms those of the original study that excluded the lasts two items from the total scoring, defining the first 15 items as “Sensory issue type 1 = Sensory over-reactivity” and the last 2 items as “Sensory under-reactivity and/or issues with perception”. Unfortunately, this means there is low content validity for the hyporeactivity dimension: the two items do not agree with the calculation of the total score. The study showed that the instrument had good internal consistency for both subscales, similar to earlier reports of internal consistency of the THPQ-R [16].

Our data showed that the 5-year-old children in our sample had lower scores on the THPQ-R, meaning they experienced more challenges related to defecation compared with the other age groups in this study or with the samples of other studies [16]. Functional constipation is common in TD children and may be due to dietary and behavioral causes or may result from events such as starting school or toilet training. In Italy, many TD 5-year-old children start primary school, an important transition that may explain the presence of more defecation problems in this age group; however, this result is not supported by other studies. Further studies with larger samples are needed to confirm this.

The current approach to treating defecation disorders is usually a combination of behavioral and medical interventions, which includes the establishment of routines and the prescription of drugs and laxatives. Occupational therapy can be considered as a complementary intervention to address underlying sensory issues impacting toileting habits [35]. Our study results support the use of the Italian version of the THPQ-R by occupational therapists to assess sensory issues impacting participation in toileting in children. The availability of psychometrically sound assessment tools is essential in understanding the potential factors linked to occupational challenges and is needed to promote clinical practice in this area of occupational therapy [36]. With the validation of this tool, it will not only be possible to define the impact of sensory problems on the acquisition of defecation habits, but also to promote cooperation between occupational therapists, pediatricians, and gastroenterologists in the treatment of functional gastrointestinal disorders, such as constipation and nonretentive fecal incontinence, linked to sensory integration issues. In addition to these clinical advantages, it will now be possible to compare the results obtained from studies conducted in Italy with those conducted in English- or Spanish-speaking countries, contributing to research in this area. This study has some limitations. The content validity is low for the hyporeactivity dimension, and the two items do not concur to the calculation of the total score. In future studies, the instrument should be analyzed for criterion, discriminant, and convergent validity. Moreover, it should be studied in developmentally impaired children, with and without defecation disorders.

## Figures and Tables

**Figure 1 children-09-01052-f001:**
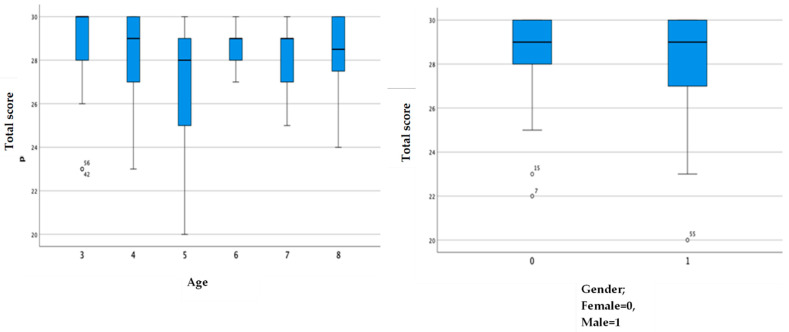
Cross cultural validity: Student’s *t*-test between and demographic characteristics of participants.

**Table 1 children-09-01052-t001:** Demographic characteristics of the population: information on the 90 participants included.

	Number	Percentage (%)
Age (years)		
3	18	20
4	17	19
5	21	23
6	9	10
7	9	10
8	16	18
Sex		
Female	39	43
Male	51	57

**Table 2 children-09-01052-t002:** Construct validity: exploratory factor analysis (EFA).

Item	Mean	Standard Deviation	Factor 1 (Sensory Over-Reactivity)	Factor 2 (Sensory Under-Reactivity)
1	1.94	0.230	0.799	
2	1.98	0.148	0.784	
3	1.92	0.269	0.817	
4	1.92	0.269	0.716	
5	1.91	0.286	0.841	
6	1.79	0.410	0.729	
7	1.97	0.181	0.529	
8	1.91	0.286	0.733	
9	1.77	0.425	0.658	
10	1.86	0.354	0.521	
11	1.90	0.302	0.519	
12	1.98	0.148	0.723	
13	1.64	0.481	0.371	
14	1.79	0.410	0.480	
15	1.84	0.364	0.373	
16	1.88	0.329		0.626
17	1.93	0.251		0.648
Variance			57.6%	6.4%

**Table 3 children-09-01052-t003:** Internal consistency: Cronbach’s alpha for the subscales and Cronbach’s alpha if item deleted.

Item	Mean	Standard Deviation	Cronbach’s Alpha If the Element Is Deleted
My child hides that they poop.	1.94	0.230	0.708
My child asks for a diaper when they feel the need to poop.	1.98	0.148	0.706
My child prefers to poop in their clothing although the potty or toilet is nearby.	1.92	0.269	0.702
My child refuses to sit on the potty or the toilet to poop, but will pee in the potty or toilet.	1.92	0.269	0.688
My child refuses or seems uncomfortable sitting on the toilet or potty for both peeing and pooping, even at home.	1.91	0.286	0.684
My child withholds poop or resists the urge to poop.	1.79	0.410	0.659
My child follows an unusual ritual when pooping, which involves actions or places not typically associated with pooping or with the age of the child.	1.97	0.181	0.701
My child seems to feel pain when pooping, even if the poop is soft.	1.91	0.286	0.693
My child refuses to poop outside of the home.	1.77	0.425	0.698
My child shows exaggerated disgust at the smell of their poop.	1.86	0.354	0.728
My child refuses to wipe or be wiped after pooping.	1.90	0.302	0.740
My child shows fear or refusal related to certain features of the bathroom, such as fear of flushing the toilet.	1.98	0.148	0.727
My child needs to pay attention to something else while pooping (a book, a game); this seems to help them tolerate the sensation of pooping.	1.64	0.481	0.714
My child is sensitive to taste and/or food textures, making it difficult to accept laxative medicine or high-fibre foods.	1.79	0.410	0.687
My child started to feel the urge to poop from a very young age (before 12 months). When my child complained in a certain way they were put on the potty to poop.	1.84	0.364	0.719
Total	1.93	0.251	0.719

## Data Availability

The data that supports the findings of this study are available from the corresponding author upon reasonable request.

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
