# Peer review of "Italian Translation, Cultural Adaptation, and Validation of the Toileting Habit Profile Questionnaire Revised (THPQ-R) in Typically Developing Children: A Cross-Sectional Study"

_children, 2022, doi:10.3390/children9071052_

Round 1
Reviewer 1 Report
The purpose of this study was to translate THPQ-R from English to Italian and to identify the validity of the translated THPQ-R tool. Even if the original tool is validated, Exploratory Factor Analysis, Confirmatory Factor Analysis must be performed in this study. In addition, the results analysis seemed to be somewhat insufficient as a study to test validity such as criterion validity, content and discriminant, convergent validity.
Since this script was in English and the readers also spoke English, please correct the Italian in the tools into English in Table 2.
Author Response
Dear Reviewer,
We appreciate the opportunity to resubmit our article entitled “Italian translation, cultural adaptation, and validation of the Toileting Habit Profile Questionnaire Revised (THPQ-R) in typically developing children: a cross-sectional study.” We would like to thank the referees for the careful and constructive reviews. We have made corresponding changes directly to the manuscript where appropriate with changes tracked. The revised version of our manuscript accompanies this letter. All comments by the reviewer have been addressed. Based on his/her comments, we have made changes to the manuscript, which are detailed below.

Reviewer 2 Report
General comments
The authors propose the translation, cultural adaptation and validation of The Toileting Habit Profile Questionnaire Revised (THPQ-R), an instrument built to identify sensory integration problems in children with defecation problems. The main contribution is the translation and adaptation of the instrument into Italian.
It is not very clear what the relevance of this instrument is over other instruments or strategies for evaluating sensory integration, nor is the conceptual definition of this construct explicitly explained. There are gaps in the procedures for cultural adaptation of the instrument. To adapt the version translated into Italian, it is necessary to identify if the people to whom the test is addressed (in this case parents of children between 3 and 8 years old), understand it and if there are equivalent terms or definitions in both cultures (English and Italian). No major analyses are carried out that give evidence of validity: Content of the instrument, internal structure, response processes, consequences of the measurement. Ethical considerations are not made explicit and there is no evidence of having submitted the protocol to an ethics committee. There are flaws in the editing of the document (for example, on page 3 there is a crossed out paragraph; in table 2 there are terms in English and Italian, margin problems).
Author Response

(The authors gave the same response as above.)

Round 2
Reviewer 1 Report
1. Please suggest based on the relevant reference on how much factor loading was adopted for each question.
2. EFA extracted 17 item that explained the 63.9% of the variance, which was contained in two major factors. In Table 2, Factor 1 was 57.6% and Factor 2 was 6.4%, which was 64.0%, which was presented in the text as 63.9%. It is good to match the values in the table and text.
3. A more in-depth discussion was needed based on the derived research results.
- For boys, at the age of 5, did the large variation in the score and the high and low average of each item appear as similar problems in other studies? etc.
- Limitations for not testing validity such as criterion validity, content and discriminant, and convergent validity should be added.
Author Response
Date: June 26th, 2022
Dear Editor,
We appreciate the opportunity to resubmit our article entitled “Italian translation, cultural adaptation, and validation of the Toileting Habit Profile Questionnaire Revised (THPQ-R) in typically developing children: a cross-sectional study.” We would like to thank the referees for the careful and constructive reviews. We have made corresponding changes directly to the manuscript where appropriate with changes tracked. The revised version of our manuscript accompanies this letter. All comments by the reviewer have been addressed. Based on his/her comments, we have made changes to the manuscript, which are detailed below.
|
Reviewer Comment |
Response |
Line # |
|
Reviewer #1 |
||
|
The purpose of this study was to translate THPQ-R from English to Italian and to identify the validity of the translated THPQ-R tool. Even if the original tool is validated, Exploratory Factor Analysis, Confirmatory Factor Analysis must be performed in this study. In addition, the results analysis seemed to be somewhat insufficient as a study to test validity such as criterion validity, content and discriminant, convergent validity. |
Exploratory Factor Analysis has been performed and the text has been modify accordingly |
18,19 118-120 157-166 182-188 |
|
Since this script was in English and the readers also spoke English, please correct the Italian in the tools into English in Table 2. |
Table 2 has been corrected |
Table 3 |
|
Reviewer #2 |
||
|
It is not very clear what the relevance of this instrument is over other instruments or strategies for evaluating sensory integration, nor is the conceptual definition of this construct explicitly explained. |
“The Toileting Habit Profile Questionnaire Revised (THPQ-R) (8,9) was designed to screen for sensory problems related to defecation disorders.” To our knowledge this is the only instrument designed for this purpose. Other instruments used in the evaluation of sensory integration issues do not contemplate defecation and toileting sensations specifically and some important information may be missed with other tools; the THPQ-R is unique in its focus on the sensory aspects of toileting.
The first version of the THPQ (7) used a five point response scale and included 11 items divided into two sections: hyperreactivity (9 items) and hyporeactivity (2 items). Changes were made to the questionnaire following studies to verify the validity of the construct (Rash analysis). The most recent version of the THPQ, the THPQ-R (9) is com-posed of 17 items and the scale is now dichotomous, with two response options for each item: 1) frequently or always or 2) never or rarely, based on the frequency with which the behavior described in the item occurs. Items 16 and 17 are not included in the calcu-lation of the final score because they were showed to represent a different construct, sensory hyporeactivity or difficulties in perception, a construct which is difficult to assess with questionnaires (9). |
59,60 101-110 |
|
There are gaps in the procedures for cultural adaptation of the instrument. To adapt the version translated into Italian, it is necessary to identify if the people to whom the test is addressed (in this case parents of children between 3 and 8 years old), understand it and if there are equivalent terms or definitions in both cultures (English and Italian). |
The final version was also been shown to some parents (children between 3 and 8 years old), who were not familiar with sensory integration terminology, to check under-standability.
A group people of the target population (parents children between 3 and 8 years old), confirmed understandability of the questionnaire. |
92-94 135-137 |
|
No major analyses are carried out that give evidence of validity: Content of the instrument, internal structure, response processes, consequences of the measurement. |
In addition to Internal consistency and cross cultural validity, exploratory factor analysis has been performed to check construct validity, according to COSMIN checklist |
18,19 118-120 157-166 182-188 |
|
Ethical considerations are not made explicit and there is no evidence of having submitted the protocol to an ethics committee. |
Ethical committee has been added |
|
|
There are flaws in the editing of the document (for example, on page 3 there is a crossed out paragraph; in table 2 there are terms in English and Italian, margin problems). |
Editing of the manuscript has been corrected |
Throughout the text |
|
Editor |
||
|
Additionally to the comments you received from reviewers, we would like you to revise your manuscript, since we have found some repetitions. You can find attached the report to see which parts on your text should be revised. We would be grateful if you tried to reword the paragraphs highlighted. |
Repetitions have been removed |
Throughout the text |
|
2. Please add at the end of your manuscript the Authors' Contribution part, where you briefly describe each authors' role in the work. |
Authors contribution has been added |
227-229 |
|
3. The length of the present version is a little shorter than what we expected for article paper. In order to increase the readability of the article and to have a deeper understanding of the research content for readers, we are kindly suggesting you to add more details or references to support your research results when you revise your manuscript. |
Some text has been added |
Throughout the text |
We hope that the new version of our manuscript is acceptable for publication.
Best regards,
Anna Berardi

Reviewer 2 Report
June 15th, 2022
Reviewer 2.
I would like to thank the authors for the effort made to address my comments. I am pleased with the answer given to some of them, although I still have some major concerns about how the authors approached the reviewer’s suggestions concerning the following:
- Regarding the AFE table, it is recommended to preserve the items with a factorial load equal to or superior to 0.40, there are two items with a lower load. The factor loadings indicate the weight of the item in each factor.
- No confirmatory factor analysis was carried out. There is no evidence of convergent or divergent validity.
- Why did you assess cross-cultural validity through the distribution of results by age and sex (boxplot)?
- It is not very clear what the relevance of this instrument is over other instruments or strategies for evaluating sensory integration, nor is the conceptual definition of this construct explicitly explained.
- A definition of “sensory problems related to defecation disorders” is required, explain how it is conceptually understood and defined. Likewise, detail the definition of hyper and hyporeactivity. What are the two dimensions that make up the instrument? These definitions are requested given the importance of reflecting the underlying theory of the phenomenon or concept to be measured (construct validity).
- The evaluation of a dimension with only two items is not highly recommended. For a factor to be considered as such on a scale, it must include at least three items; with two, it is only an indicator
Reference: Michael Furr, 2021, Psychometrics: An Introduction, SAGE: Chapter 4. Test Dimensionality and Factor Analysis.
Therefore, there is low content validity for the hyporeactivity dimension.
- It is not explained how the application process (online and face-to-face) could have impacted validity. For the online application, it was probably required to adapt the instrument and its instructions to the application format, this is not detailed. The process of recruiting the sample is not explained as well. It is necessary to describe to whom (for example, from which regions of Italy), and how the invitation was made (if the call was published on social networks, in schools, etc).
- Is the online or the paper version validated?
- The sample size is small. It is recommended to increase it.
- The research design is not clear, it is not understood the data collection process. Is the research primary or secondary?
- No ethics approval from a committee was added.
Reviewer General Comments
Authors need to reflect on the following:
Why is it relevant to have an Italian version of this instrument, beyond making comparisons with other studies? I am afraid there is no greater scientific merit in translating, adapting, and validating the instrument in a limited and incomplete manner.
What version of the instrument are you validating, the original or the revised one? Given that the data collection strategy was done online and on paper, are you validating the online version of the instrument, the one on paper, or both? And why not make a comparison or differentiation? While we expect perhaps few overall differences between the formats, it does have implications for instrument fit and validity. For example, it can be argued that differences could arise to the extent that participants feel more comfortable and willing to answer honestly in an online format, which provides a degree of anonymity. It is recommended to carry out a second survey to perform the confirmatory factor analysis and, based on the nomological network of the construct being evaluated, to be able to identify its relationship with other related constructs and to test convergent and divergent validity.

Author Response

(The authors gave the same response as above.)

Round 3
Reviewer 1 Report
Thank you for your sincere reply to the comment on the review.
By presenting the unanalyzed part as a limitation in the manuscript, other researchers will be able to gain insight into future research through this study.
Reviewer 2 Report
Many thanks to the authors for the effort made in this version of the manuscript.
Following the recommendations of the Committee on Publication Ethics, the authors must offer the details of ethical approval and informed consent for studies in humans. Especially when the study population are minors.
I'm sorry for insisting on the ethical considerations: the researchers carried out the primary study with patient recruitment. They have a sample of 118 when the parents were only 90, how is this explained? For the participation of all minors, they should have obtained the consent of all parents. A parent for each child.
No ethics approval from a committee was added.
Committee on publication ethics (COPE). A Short Guide to Ethical Editing for New Editors. Version 2, 2016. Available in: https://publicationethics.org/files/A_Short_Guide_to_Ethical_Editing.pdf